# Effects of Amnioreduction before Physical Examination-Indicated Cerclage on Pregnancy Outcomes: A Propensity Score Matched Study [note 1]

**DOI:** 10.3390/jcm12072480

**Published:** 2023-03-24

**Authors:** Subeen Hong, Hyun Sun Ko, Seonok Kim, Yun Sung Jo, In Yang Park

**Affiliations:** 1Department of Obstetrics and Gynecology, Seoul St. Mary’s Hospital, College of Medicine, The Catholic University of Korea, Seoul 06591, Republic of Korea; 2Department of Clinical Epidemiology and Biostatistics, ASAN Medical Center, Seoul 05505, Republic of Korea; 3Department of Obstetrics and Gynecology, St. Vincent’s Hospital, College of Medicine, The Catholic University of Korea, Suwon 16247, Republic of Korea

**Keywords:** amnioreduction, physical examination-indicated cerclage, pregnancy outcome, premature birth, propensity score

## Abstract

This study investigated the effects of amnioreduction before physical examination-indicated cerclage on pregnancy outcomes using a propensity score matching analysis. This multicenter retrospective cohort study included women who underwent cerclage operations due to painless cervical dilation in the second trimester (14–28 weeks). The primary outcome was the time from operation until delivery. Secondary outcomes included preterm birth rate and neonatal outcomes. Primary and secondary outcomes were compared between those with amnioreduction and those without amnioreduction. Of 103 women, 31 received preoperative amnioreduction (amnioreduction group) and 72 women did not (no-amnioreduction group). Since there were differences in baseline characteristics and preoperative ultrasound findings between the two groups, we matched 25 women with amnioreduction and 25 women without amnioreduction using a propensity score. In the matched cohort, the amnioreduction group showed a shorter time from operation to delivery than the group without amnioreduction and the hazard ratio of amnioreduction was 2.5 (95% confidence interval; 1.4–4.7). In addition, the preterm birth rate before 28 weeks of gestation and the neonatal composite outcome were higher in the amnioreduction group than that in the group without amnioreduction. Amnioreduction before physical examination-indicated cerclage was associated with poor pregnancy and neonatal outcomes. Therefore, careful consideration is required when performing amnioreduction before cerclage operation.

## 1. Introduction

Cervical incompetence has been characterized classically by painless cervical dilatation in the mid-trimester, which is a major cause of extremely preterm birth or pregnancy loss. When there is a cervical dilatation with a visible amniotic membrane on physical examination, cerclage placement is necessary to reinforce the cervix. This is called physical examination-indicated cerclage [1,2]. It has been reported that women who have undergone physical examination-indicated cerclage show superior perinatal outcomes than those who are managed expectantly [3,4]. According to a systematic review, neonatal survival improved by 1.6 times and decreased by 0.23 times in delivery at 24–28 weeks when cerclage was performed [4].

However, cerclage operation is technically challenging in cases of cervical dilatation. Intraoperative rupture of membranes and cervical laceration are complications of emergent cerclage operation, having an incidence of 4–8% [4,5,6]. In particular, the difficulty of operation is increased as cervical stitching is performed while replacing the protruding membrane into the cervix.

Several methods have been introduced to reposition the amniotic membrane into the uterus. Pulling the cervix with ring forceps or pushing the bulging bag with saline gauze or foley catheter balloons has been used [2,7,8,9]. Among those methods, amnioreduction facilitates the replacement of the protruding membrane by reducing the hydrostatic force. Through several studies, amnioreduction has been proven to be a feasible method [10,11,12,13,14,15].

Although it is important to evaluate whether this procedure is really beneficial when performing cerclage, there is a paucity of information regarding whether amnioreduction actually improves pregnancy and neonatal outcomes compared to other methods in the context of physical examination-indicated cerclage. Thus, the purpose of this study was to investigate the effect of amnioreduction on pregnancy and neonatal outcomes in women who underwent physical examination-indicated cerclage.

## 2. Methods

### 2.1. Study Design and Study Population

This retrospective cohort study was conducted in two centers (Seoul St. Mary’s Hospital and St. Vincent’s Hospital) in South Korea. We collected data of women who underwent physical examination-indicated cerclage due to painless cervical dilatation between 14–28 weeks of gestation from January 2009 to December 2020. Women with ruptured membranes or multiple pregnancies were excluded. To examine the efficacy of amnioreduction, the study population was divided into two groups according to whether or not amnioreduction was performed before surgery. Delivery outcomes of the two groups were then compared using a propensity score matching analysis. We have obtained the approval for this study from the Central Institutional Review Board of the Catholic Medical Center in South Korea.

### 2.2. Determination of Amnioreduction and Physical Examination-Indicated Cerclage

The examination was performed for women who had symptoms such as vaginal bleeding or discharge and those with amniotic membrane protrusion who were referred from another hospital. If a protruding membrane was confirmed by examination, McDonald operation after replacing the amniotic membrane into the uterus was planned. The decision to carry out amnioreduction was determined based on the severity of the protrusion of the amniotic sac and the doctor’s preference. Amnioreduction was performed using a 20–22-gauge needle under ultrasound-guided conditions. Amniotic fluid was removed as much as possible or until the protruding sac decreased upon cervical examination. In this study, the median amounts of removed amniotic fluid were 175 mL (range: 50–400 mL). Amnioreduction was not performed if amniocentesis was not agreed on or if amniocentesis could not be performed. In most cases, prophylactic tocolytics and antibiotics were used. Regardless of intraoperative amnioreduction, the amnion was replaced by pulling the cervix with sponge forceps and pushing the amniotic membrane gently with a saline gauze during the operation.

### 2.3. Measurement of Protruding Amniotic Sac

Ultrasound parameters in ultrasound scans taken before surgery were measured retrospectively by two physicians (S Hong and YS Jo). Amniotic sludge, the shape of the protruding sac, width and length of the protruding sac, funneling length and width, and width of the narrowest point of the cervix were measured. The measurement method is shown in Appendix A. In this figure, A is the maximal width of the protruding sac, B is the maximal length of protruding sac, C is the length of funneling, D is the width of funneling, and E represents the width of the narrowest point of the cervix.

### 2.4. Definitions of Outcomes

The primary outcome was the interval from operation to delivery. Secondary outcomes were operation failure, preterm birth before 24, 28, and 34 weeks of gestation, and neonatal outcomes. Operation failure was defined as a case when the operation was not possible because the amnion was not replaced during the operation and when the membrane was ruptured at the time of surgery or within 24 h of surgery. Neonatal composite outcomes included respiratory distress syndrome, bronchopulmonary dysplasia, necrotizing enterocolitis, intraventricular hemorrhage, sepsis, and neonatal death.

### 2.5. Statistical Analysis

For analysis of the total study population, the Student’s *t*-test or Wilcoxon rank–sum test was used for comparing continuous variables, and the Chi-square test or Fisher’s exact test was used for comparing categorical variables between the two groups. To reduce the influence of potential covariates or selection bias, the independent biostatistician (S Kim) performed propensity score matching to create maximally comparable groups. The possible covariates included for propensity score were the following: maternal demographics (age, parity, pre-pregnancy body mass index, and history of preterm birth), characteristics at admission (gestational age at admission, laboratory findings on admission, symptoms, results of vaginal culture, cervical dilatation, use of tocolytics, and use of antibiotics), and ultrasound parameters (amniotic sludge, appearance of protruding sac, width and length of protruding sac, width and length of funneling, and width of narrowest point of the cervix).

A ‘greedy’-matching algorithm was performed to match subjects using a caliper of 0.2 standard deviations of the logit of the propensity score and the standardized mean difference (SMD) was used for assessing the balance of variables before and after matching. After matching, a paired t-test or Wilcoxon signed–rank test was used to compare continuous variables, and a McNemar’s test was used to compare categorical variables between the two groups. The risk of neonatal outcomes was compared with the use of logistic models using Generalized Estimating Equations (GEE) that accounted for the clustering of matched pairs. It is reported as odds ratios (OR) and 95% confidence intervals (CI). Maternal outcomes and survival outcomes were compared using logistic models and Cox models, respectively, with robust standard errors. They are reported as OR or hazard ratio (HR) and 95% CI, as appropriate. All statistical analyses were performed with SAS version 9.4 (SAS Institute, Cary, NC, USA) or R version 3.6.1. Two-tailed *p*-values < 0.05 were considered statistically significant.

## 3. Results

Table 1 shows baseline characteristics and ultrasound parameters of the study population before and after propensity score matching. In the total study population, the amnioreduction group less incidentally found showed more dilated cervix rates upon admission (accidental discovery, *p* < 0.05; cervical dilatation > 2 cm, *p* < 0.005). Among ultrasound parameters, hour glassing appearance was observed more in the amnioreduction group (*p* = 0.001). There were significant differences in the width and length of the protruding sac between the two groups (width of protruding sac: mean 3.1 cm vs. 1.9 cm, *p* = 0.005; length of protruding sac: mean 2.0 cm vs. 1.0 cm, *p* < 0.001). After propensity score matching to correct these differences, there were no differences in maternal demographics, characteristics at admission, or ultrasound parameters (all *p*-value > 0.5, all SMD < 0.2).

Table 2 demonstrates obstetric outcomes between the amnioreduction group and the no-amnioreduction group after matching. The amnioreduction group showed earlier gestational age at delivery than the no-amnioreduction group, although their difference was not statistically significant (mean: 24.8 weeks vs. 28.5 weeks, *p* = 0.051). The mean interval from operation to delivery of the amnioreduction group was 3.1 weeks of gestation, which was shorter than that of the no-amnioreduction group (mean: 3.1 weeks vs. 6.8 weeks, *p* < 0.05). The probability of delivery was significantly higher in the amnioreduction group both before and after matching, as presented in Figure 1 (HR before matching: 3.22, 95% CI: 1.97–5.27, *p* < 0.001; HR after matching: 2.50, 95% CI: 1.35–4.65, *p* < 0.005). The rate of operation failure and preterm birth before 24 weeks of gestation were not significantly different between the two groups. However, rates of preterm birth before 28 and 34 weeks of gestation of the amnioreduction group were higher than those of the no-amnioreduction group (preterm birth <28 weeks: 80% vs. 54%, OR: 3.39, 95% CI: 1.05–10.94, *p* < 0.05; preterm birth <34 weeks: 92% vs. 70%, OR: 5.03, 95% CI: 1.02–24.77, *p* < 0.05).

Table 3 shows neonatal outcomes of living fetuses of the matched population. NICU admission and composite morbidity of the amnioreduction group were higher than those of the no-amnioreduction group (NICU admission, OR: 5.50, 95% CI: 1.09–27.75, *p* < 0.05; composite morbidity, OR: 7.33, 95% CI: 1.47–36.67, *p* < 0.05).

## 4. Discussion

In this study, it was found that amnioreduction was performed in more severe patients. When the two groups were analyzed after propensity score matching, amnioreduction did not show any benefit in pregnancy prolongation. In women who received amnioreduction, the risk of preterm birth <28 weeks and <34 weeks of gestation and composite neonatal outcomes were higher than those who did not.

The protruding membrane in women with cervical incompetence not only suggests a poor prognosis, but also requires complicated surgical techniques to reposition the membrane into the uterus [16]. Although several methods have been introduced to reposition the membrane, there is no evidence that any method is superior [2,17].

Amnioreduction was originally attempted for symptom reduction and better pregnancy outcomes in twin-to-twin transfusion syndrome or idiopathic polyhydramnios. It was introduced as one of facilitating methods of bag repositioning by reducing the intra-amniotic pressure in 1980 [13,18]. Afterwards, several studies reported the feasibility of amnioreduction. They are summarized in Table 4. Interestingly, studies comparing the amnioreduction group and no-amnioreduction group reported to date all showed different results.

Lacatelli et al. reported the superiority of amnioreduction for a better pregnancy outcome [12]. The rate of preterm birth <32 weeks was significantly less (1/7 vs. 6/8) and the duration of neonatal hospital stay was shorter (3 vs. 37 days) in women with amnioreduction compared to those without amnioreduction.

Cakroglu et al. reported no difference in pregnancy or neonatal outcomes according to amnioreduction [11]. The authors argued that amnioreduction was meaningful as a means to determine the presence of chorioamnionitis even though it did not improve the prognosis.

Finally, Makino et al. suggested that amnioreduction was a useful technique for repositioning fetal membranes, although it had the risk of rupture of membranes [10]. In the present study, 3 out of 11 patients who underwent amnioreduction showed rupture at the time of surgery. Except for these cases, there was no significant difference in pregnancy prolongation between the two groups (32.9 days vs. 36.9 days, *p* = 1.000), although the amnioreduction group had more severe cervical dilation than the no amnioreduction group (6.7 cm vs. 4.1 cm, *p* < 0.005).

In this study, there were differences in the severity of cervical incompetence between the group with and without amnioreduction. Thus, a propensity score match was performed to eliminate selection bias. As a result, the time from surgery to delivery was shorter and the rate of preterm birth and the risk of composite neonatal outcome were higher in women who received amnioreduction than in those who did not. The risk of rupture of the membrane within 24 h or inoperability was higher in the amnioreduction group, although the difference between the two groups was not statistically different.

There are several possibilities for the higher risk of preterm birth in women who have received amnioreduction before physical examination-indicated cerclage.

First, there is potential for triggering preterm parturition by an aggravation of intraamniotic infection and inflammation. In women with cervical incompetence, up to 80% have intraamniotic infection and inflammation [19,20,21]. The amniotic barrier is already weakened under inflammatory cytokines [22,23]. Puncture of membranes can aggravate the weakness of the membrane because it destroys the continuity of membranes and provides a pathway for microorganism entry into the amniotic fluid [24].

Second, iatrogenic disruption or weakening adherence between uterine decidua and fetal membranes may occur by removing a large amount of amniotic fluid. There are some traditional concerns about the rupture of membranes or placenta abruption resulting from amnioreduction, especially through the lower uterine segment [25]. In addition, chorioamnionic separation may occur due to amniotic fluid efflux into the chorioamnionic space after an invasive procedure, which is associated with a poor pregnancy outcome [26,27,28].

To the best of our knowledge, our study included the largest number of patients among studies comparing methods to replace the protruding membranes in the uterus. In addition, to reduce selection bias due to its retrospective nature, the amnioreduction effect was verified through case-control propensity score matching analysis. In particular, ultrasound findings, which were the most important objective findings on prognosis, were reviewed by two physicians and as many variables as possible were matched to make the two groups comparable.

Despite the above efforts, selection bias could exist and the number of patients was too small to evaluate neonatal outcomes. Since amniocentesis was not performed in the control group, we could not compare intraamniotic infection or inflammation. A large-scale prospective study comparing the methods of replacing the membranes is needed. Information on various biological markers as well as ultrasound findings is required in further studies.

In conclusion, amnioreduction as a method of repositioning the protruding membranes did not improve pregnancy prognosis, but rather worsened pregnancy outcomes. Therefore, it is necessary to try other methods first for amnion repositioning. Amnioreduction should be selectively performed for patients who cannot proceed with other methods.

## Figures and Tables

**Figure 1 jcm-12-02480-f001:**
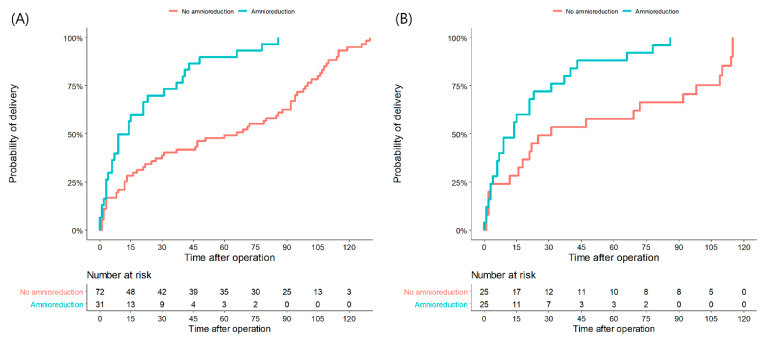
Kaplan–Meier curves for time to delivery in unmatched(**A**) and matched patients (**B**) who underwent amnioreduction or did not. Survival curves show the differences in delivery proportions according to the latency period (days from operation to delivery) between two groups.

**Table 1 jcm-12-02480-t001:** Baseline characteristics and ultrasound parameters of the eligible study population and propensity score-matched population.

		Eligible Study Population (*n* = 104)				Propensity Score Matched Population (*n* = 50)		
	Amnioreduction (*n* = 31)	No Amnioreduction (*n* = 72)	*p*-Value	SMD	Amnioreduction (*n* = 25)	No Amnioreduction (*n* = 25)	*p*-Value	SMD
		Maternal Demographics						
Age (year)	33.5 ± 4.6	32.4 ± 3.8	0.631	0.248	33.5 ± 4.6	33.1 ± 3.9	0.814	0.103
Nulliparity	16 (51.6%)	36 (50.0%)	0.881	0.032	12 (48.0%)	10 (40.0%)	0.774	0.162
History of preterm birth	4 (12.9%)	6 (8.3%)	0.483	0.149	2 (8.0%)	2 (8.0%)	1.000	0
Prepregnancy BMI (kg/m^2^)	24.0 ± 3.2	23.4 ± 4.5	0.231	0.170	24.1 ± 3.3	24.3 ± 3.8	0.862	0.050
		Characteristics at Admission						
GA at admission (weeks)	21.5 ± 2.4	22.0 ± 2.8	0.231	0.216	21.5 ± 2.6	21.5 ± 2.6	0.969	0.027
GA at operation (weeks)	21.6 ± 2.3	22.2 ± 2.9	0.242	0.213	21.6 ± 2.5	21.5 ± 2.6	0.925	0.038
Serum WBC (cells/mL)	10,753 ± 2256	10,790 ± 3114	0.765	0.013	11,024 ± 2395	10,936 ± 2822	0.903	0.034
Serum CRP (mg/dL)	0.82 ± 0.88	0.92 ± 1.35	0.749	0.090	0.84 ± 0.87	0.74 ± 0.92	0.855	0.117
Accidental discovery ^a^	10 (32.3%)	40 (55.6%)	0.030	0.483	9 (36.0%)	9 (36.0%)	1.000	0
Positive vaginal culture for bacteria ^b^	8/29 (27.6%)	17/70 (24.3%)	0.731	0.075	8/24 (33.3%)	8 (32.0%)	1.000	0.028
Positive vaginal culture for ureaplasma species ^b^	14/29 (48.3%)	26/70 (37.1%)	0.304	0.227	10/24 (41.7)	9 (36.0%)	1.000	0.116
Cervical dilatation > 2 cm ^c^	31 (100%)	44/62 (73.3%)	0.002	0.853	25 (100%)	23/23 (100%)	-	0
Use of tocolytics	29 (93.6%)	70 (97.2%)	0.582	0.176	24 (96.0%)	24 (96.0%)	1.000	0
Use of antibiotics	2 (6.5%)	0 (0%)	0.089	0.371	25 (100%)	25 (100%)	-	0
		Ultrasound Parameters ^b^						
Amniotic sludge	20/29 (69.0%)	29/62 (46.8%)	0.001	0.445	15/24 (62.5%)	15/24 (62.4%)	0.688	0
Hour glassing appearance of protruding sac	23/29 (79.3%)	26/62 (41.9%)	0.056	0.828	18/24 (75.0%)	17/24 (70.8%)	1.000	0.094
Width of protruding sac (cm)	3.1 ± 1.7	1.9 ± 1.8	0.005	0.749	3.1 ± 1.6	3.0 ± 1.7	0.873	0.044
Length of protruding sac (cm)	2.0 ± 1.4	1.0 ± 1.3	<0.001	0.760	1.9 ± 1.6	2.0 ± 1.4	0.654	0.077
Length of funneling (cm)	3.5 ± 0.8	3.7 ± 1.2	0.361	0.194	3.6 ± 1.0	3.7 ± 0.8	0.742	0.050
Width of Funneling (cm)	2.7 ± 0.8	2.6 ± 1.1	0.855	0.043	2.6 ± 1.0	2.6 ± 0.8	0.862	0.018
Width of narrowest point of the cervix (cm)	1.2 ± 0.8	1.2 ± 0.7	0.623	0.060	1.1 ± 0.8	1.2 ± 0.7	0.807	0.111

Data are presented as mean ± 2 standard error or *n* (%). SMD, standardized mean difference; BMI, body mass index; GA, gestational age; WBC, white blood cell count; CRP, C-reactive protein. ^a^ Accidental discovery means the cases without symptoms, such as vaginal bleeding or discharge, and accidentally discovered at their routine antenatal visit. ^b^ Vaginal culture tests were not performed for four patients. ^c^ Ultrasound results were not recorded for twelve patients. Among them, the degree of cervical dilatation on the speculum exam was described in two patients.

**Table 2 jcm-12-02480-t002:** Obstetric outcomes of propensity score-matched population.

	Amnioreduction (*n* = 25)	No Amnioreduction (*n* = 25)	Hazard Ratio	95% CI
GA at delivery (weeks) ^a^	24.8 ± 5.2	28.5 ± 7.4	-	-
Interval from operation to delivery (weeks) ^a^	3.1 ± 3.5	6.8 ± 6.4	2.50	1.35–4.65
			Odds ratio	95% CI
Operation failure	6 (24.0%)	3 (12.0)	2.32	0.54–9.93
Preterm birth <24 weeks ^b^	14/20 (70.0%)	9/22 (40.9%)	3.37	0.94–12.12
Preterm birth <28 weeks ^c^	20 (80.0%)	13/24 (54.2%)	3.39	1.05–10.94
Preterm birth <34 weeks ^d^	23 (92.0%)	16/23 (69.6%)	5.03	1.02–24.77
Delivered living fetus ^d^	13 (52.0%)	14/23 (60.9%)	0.70	0.26–1.83

Data are presented as mean ± 2 standard error or *n* (%). GA, gestational age; CI, confidence interval. ^a^ For six patients who were not available with delivery records, gestational age at delivery was calculated as the date of the last follow-up. ^b^ Seven patients who were hospitalized after 24 weeks of gestation and one patient who lost follow-up until 24 weeks of gestation were excluded. ^c^ One patient who lost follow-up until 28 weeks of gestation was excluded. ^d^ Two patients who lost follow-up until 34 weeks of gestation were excluded.

**Table 3 jcm-12-02480-t003:** Neonatal outcomes of living fetuses of propensity score-matched population.

	Amnioreduction (*n* = 13)	No Amnioreduction (*n* = 14)	Odds Ratio	95% CI
Composite morbidity ^a^	11 (84.6%)	6 (42.9%)	7.33	1.47–36.67
RDS	10 (76.9%)	6 (42.9%)	4.44	0.87–22.76
BPD	5 (38.5%)	3 (21.4%)	2.29	0.44–11.89
NEC	1 (7.7%)	3 (21.4%)	0.31	0.03–3.60
IVH	6 (46.2%)	3 (21.4%)	3.14	0.73–13.53
Sepsis	3 (23.1%)	3 (21.4%)	1.10	0.20–5.93
Neonatal death	1 (7.7%)	3 (21.4%)	0.31	0.02–3.96
NICU admission	11 (84.6%)	7 (50.0%)	5.50	1.09–27.75

Data are presented as *n* (%). CI, confidence interval; RDS, respiratory distress syndrome; BPD, bronchopulmonary dysplasia; NEC, necrotizing enterocolitis; IVH, intraventricular hemorrhage; NICU, neonatal intensive care unit. ^a^ Composite morbidity included RDS, BPD, NEC, IVH, sepsis, and neonatal death.

**Table 4 jcm-12-02480-t004:** Existing studies on amnioreduction before physical examination-indicated cerclage excluding case reports.

Study	Year	Study Design	Amount of Removed Amniotic Fluid		Amnioreduction					No Amnioreduction			
				Case	Cervical Dilataion (cm)	GA at Cerclage (Weeks)	GA at Delivery (Weeks)	Pregnancy Prolongation (Days)	Case	Cervical Dilataion (cm)	GA at Cerclage (Weeks)	GA at Delivery (Weeks)	Pregnancy Prolongation (Days)
Makino et al. [10]	2004	Prospective cohort study	NS	8	6.7	22.1	26.5	32.9	9	4.1	23.7	29.2	36.9
Cakiroglu et al. [11]	2016	Retrospective cohort study	10 mL/week	26	5.0	21.3	28.3	53.7	30	4.0	20.6	28.1	47.3
Locatelli et al. [12]	1999	Retrospective cohort study	220–340 mL	7	3.0	21	36	100	8	2.0	23	27	10
Goodlin et al. [13]	1979	Case series	40–150 mL	9	3–4	NS	NS	NS	-		-	-	-
Zhang et al. [14]	2020	Case series	60–230 mL	8		22.4	NS	18	-		-	-	-
Proctor et al. [15]	2021	Case series	350–600 mL	7		23.9	34.3	73	-		-	-	-
Our study	2023	Retrospective cohort study	50–400 mL	31	3.1	21.6	24.7	19	72	1.9	22.2	30.2	51
		After matching		25	3.1	21.6	24.8	22	25	3.0	21.5	28.5	48

NS, no statement; GA, gestational age.

## Data Availability

All data generated or analyzed during this study are included in this article. Further enquiries can be directed to the corresponding author.

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
