# Peer review of "Effects of Amnioreduction before Physical Examination-Indicated Cerclage on Pregnancy Outcomes: A Propensity Score Matched Study†"

_jcm, 2023, doi:10.3390/jcm12072480_

Round 1

Reviewer 1 Report

The manuscript ID: jcm-2273923, titled „Effects of Amnioreduction before Physical Examination-Indicated Cerclage on Pregnancy Outcomes: A Propensity Score Matched Study” written by Subeen Hong, Hyun Sun Ko, Seonok Kim, Yun Sung Jo and In Yang Park. The authors made important observations regarding the performance of amnioreduction as a method of repositioning the protruding membranes and the impact of this procedure on the further course of pregnancy. The obtained results are clinically significant. Amnioreduction should be selectively performed for patients who cannot proceed with other methods, as suggested by the authors.

However, there is the  point for corrections prior to publication:

In the chapter "Materials and methods" there is a mention of the supplementary figure. It would be advisable to give this photo a number and add a description.

Author Response

Response to Reviewer 1 Comments

Point 1: In the chapter "Materials and methods" there is a mention of the supplementary figure. It would be advisable to give this photo a number and add a description.

Response:

I appreciate that you raised a good point. We inserted the figure number and the description in the Materials and methods section, page 3, lines 109-112.

Revised contents:

<2.3. Measurement of protruding amniotic sac>

The measurement method is shown in supplementary figure 1. In this figure, A is the maximal width of protruding sac, B is the maximal length of protrud-ing sac, C is the length of funneling, D is the width of funneling, and E represents the width of the narrowest point of the cervix.

Reviewer 2 Report

Effects of Amnioreduction before Physical Examination-Indicated Cerclage on Pregnancy Outcomes: A Propensity Score Matched Study

It is an interesting study, I enjoyed reading it.

Research is a dynamic activity and it is gradually growing but not reaching maturity. Research is a dynamic activity and it is gradually growing but not reaching maturity. The study is well-compiled and presented. I have a few suggestions; my suggestions are not judgmental and can be challenged and subjected to override if seemed inappropriate to the author or Editor. I’ll feel pleasure if I was contacted for that.

Introduction:

Line 73-75 one sentence paragraph should be merged with the last paragraph of the introduction which is written in the form of rationale of the study.

In table 1:  Nulliparity 16 (51.6) 36 (50.0); if 51.6 and 50.0 are percentage then please write it like that Nulliparity 16 (51.6%) 36 (50.0%). There a number of other such corrections of adding “%” sign.

“Accidental discovery” of what?

Reviewer 3 Report

This is an interesting study, however I do have some concerns. The numbers of women evaluated is very small with only 25 women in each group after using propensity score because of differences between groups. Secondly on line 95 it is stated that women were chosen for amnioreduction based on the severity of protrusion of the amniotic sac. It would make logical sense that in a comparison of women with less protrusion,  the intervention group was  at much greater risk for an adverse outcome  than a group chosen with less protrusion of the membranes even without any other intervention.  In addition to having greater protrusion of the membranes an intervention is undertaken (amnioreduction) which would further increase the risk of an adverse outcome. Then with propensity scoring the groups  were matched with the outcomes being earlier gestational age at delivery (NS), the time from intervention to delivery (0.05). Also deliveries < 24 weeks were similar but were greater in the amnioreduction group at < 28 and < 34 weeks. The take home message from this study reveals > risks for adverse outcomes with amnioreduction which is logical since these women are most at risk for an adverse outcome because the protrusion of the membranes was greater and another interventions the amnioreduction was undertaken. To evaluate true benefit or harm the best method would be to compare women with similar protrusion of the membranes and randomized then to amnioreduction or not and evaluate the results. There is certainly some selection bias in this study with the most severely affected women undergoing the amnioreduction compared to the control group . 

Author Response

Point 1: The numbers of women evaluated is very small with only 25 women in each group after using propensity score because of differences between groups.

Response 1:

Since there are some differences in the severity of protrusion between the two groups, we had to use propensity score matching analysis for a comparable comparison, and only small numbers were included in the comparison. However, this study has the largest number of patients among the studies evaluating the effects of amnioreduction. In addition, comparable groups were created and compared through the matching method. We described these strengths and limitations in discussion section, page 11, lines 281-290.

Point 2:  There is certainly some selection bias in this study with the most severely affected women undergoing the amnioreduction compared to the control group . 

Response 2:

In order to avoid selection bias, it is reasonable to conduct a randomized controlled trial targeting patients with a similar degree of protrusion. However, it has not been attempted to date. Our study tried to compare the two groups by including as many variables as possible, especially ultrasound severity. Nevertheless, the possibility of selection bias is a methodological limitation of the retrospective nature of the study. This is described in discussion section, page 11, line 289.

Manuscript contents, page 11, line 281-290:

To the best of our knowledge, our study included the largest number of patients among studies comparing methods to replace the protruding mem-branes into the uterus. In addition, to reduce selection bias due to retrospective nature, the amnioreduction effect was verified through case-control propensi-ty score matching analysis. In particular, ultrasound findings, which were the most important objective findings on prognosis, were reviewed by two physi-cians and as many variables as possible were matched to make the two groups comparable.

Despite the above efforts, selection bias could exist and the number of pa-tients was too small to evaluate neonatal outcomes.

Round 2

Reviewer 3 Report

Acceptable in its present form